# Effect of BCG Revaccination on Occupationally Exposed Medical Personnel Vaccinated against SARS-CoV-2

**DOI:** 10.3390/cells10113179

**Published:** 2021-11-15

**Authors:** Espiridión Ramos-Martinez, Ramcés Falfán-Valencia, Gloria Pérez-Rubio, Warrison Athanasio Andrade, Jorge Rojas-Serrano, Enrique Ambrocio-Ortiz, Dennisse S. Galicia-Álvarez, Isaac Bárcenas-Montiel, Andrea Velasco-Medina, Guillermo Velázquez-Sámano

**Affiliations:** 1Unidad de Investigación en Medicina Experimental, Facultad de Medicina, Universidad Nacional Autónoma de México, Ciudad de México 06720, Mexico; 2HLA Laboratory, Instituto Nacional de Enfermedades Respiratorias Ismael Cosío Villegas, Mexico City 14080, Mexico; rfalfanv@iner.gob.mx (R.F.-V.); glofos@yahoo.com.mx (G.P.-R.); e_ambrocio@iner.gob.mx (E.A.-O.); 3Departamento de Biologia Celular e Molecular e Bioagentes Patogênicos, Faculdade de Medicina de Ribeirão Preto, Universidade de São Paulo, Ribeirão Preto 14049-900, SP, Brazil; warrisonbio@yahoo.com.br; 4Unidad de enfermedades del Intersticio Pulmonar y Reumatología, Instituto Nacional de Enfermedades Respiratorias, “Ismael Cosío Villegas”, Ciudad de México 14080, Mexico; jrojas@iner.gob.mx; 5Servicio de Alergia e Inmunología Clínica, Hospital General de México, “Dr. Eduardo Liceaga”, Mexico City 06720, Mexico; dennissegalicia.alvarez@hotmail.com (D.S.G.-Á.); isaacbarcenasm@gmail.com (I.B.-M.); andreavelasco@hotmail.com (A.V.-M.); gvelazquezsamano@yahoo.com (G.V.-S.)

**Keywords:** SARS-CoV-2, COVID-19, BCG, vaccination, neutralizing antibodies

## Abstract

The production of specific neutralizing antibodies by individuals is thought to be the best option for reducing the number of patients with severe COVID-19, which is the reason why multiple vaccines are currently being administered worldwide. We aimed to explore the effect of revaccination with BCG, on the response to a subsequent anti-SARS-CoV-2 vaccine, in persons occupationally exposed to COVID-19 patients. Two groups of 30 randomized participants were selected: one group received a BCG revaccination, and the other group received a placebo. Subsequently, both groups were vaccinated against SARS-CoV-2. After each round of vaccination, the serum concentration of Th1/Th2 cytokines was determined. At the end of the protocol, neutralizing antibodies were determined and the HLA-DRB loci were genotyped. The participants from the BCG group and anti-SARS-CoV-2 vaccine group had increased serum cytokine concentrations (i.e., IL-1β, IL-4, IL-6, IL-12p70, IL-13, IL-18, GM-CSF, INF-γ, and TNF-α) and higher neutralizing antibody titers, compared to the group with Placebo–anti-SARS-CoV-2. Twelve HLA-DRB1 alleles were identified in the Placebo–anti-SARS-CoV-2 group, and only nine in the group revaccinated with BCG. The DRB1*04 allele exhibited increased frequency in the Placebo–anti-SARS-CoV-2 group; however, no confounding effects were found with this allele. We conclude that revaccination with BCG synergizes with subsequent vaccination against SARS-CoV-2 in occupationally exposed personnel.

## 1. Introduction

More than 100 years after the Spanish flu pandemic (1918), the world has faced a new pandemic with a considerable impact on the health and quality of life of humanity [1]. In early December 2019, the first cases of pneumonia of unknown origin were identified in Wuhan, the capital of Hubei province, China [2]. Later, the pathogen was identified as a novel coronavirus beta-enveloped RNA (SARS-CoV-2) [3]; this new pathogen clinically manifests as a severe acute respiratory syndrome (SARS) caused by coronavirus 2, which has a phylogenetic similarity to SARS-CoV [4]. Due to the high prevalence and extended incubation periods, where infected persons frequently show no symptoms, SARS-CoV-2 rapidly spread and has infected millions of people worldwide, reaching the level of a pandemic [5]. The transmission of the virus has been documented to be primarily through direct contact or respiratory droplets [6], in a proximity- and time-dependent manner, often requiring close contact within 6 feet, over 15 min or longer [7]. The process of viral transmission is complex and governed by a range of factors not fully understood; in combination, this determines the likelihood of successful infection and onward spread [8]. The first barrier that viruses must overcome to infect a new host is entry into the host cell, a phenomenon in which two variables intervene: the first is efficient virus binding to the host cell—in the case of SARS-CoV-2, this is due to its high affinity for the ACE2 receptor; and the second is the host-mediated inhibition of this process, through mechanisms such as neutralization by specific antibodies [9]. The production of these types of antibodies has been suggested to be the most efficient way to reduce the impact of this virus on the global population. Preventative vaccination is the safest and most cost-effective way to prevent COVID-19 illness and death, and the best option to combat anticipated future variants [5]. Multiple vaccination programs were initiated in the second part of 2020, using vaccines obtained with different technologies approved by the World Health Organization; these included some based on messenger RNA from the Spike protein, some used the inactivated SARS-CoV-2 virus, and some utilized viral vectors expressing the Spike protein [10]. Although all vaccines share the same goal, i.e., to induce adequate activation of the immune system, which culminates in an effective immunological memory, unfortunately, the immune response to a pathogen is often heterogeneous and varies between individuals based on age, the environment, and underlying health conditions, such as overt or silent comorbidities [11].

On the other hand, the Bacillus Calmette–Guérin (BCG) vaccine has been used since 1921, initially for protection against tuberculosis [12]. However, it has been documented that BCG acts through the stimulation of cell-mediated adaptive immunity with the activation of Th1 cells and the production of cytokines [13]; this kind of regulation has been proposed to attenuate different pathologies [14,15]. Prior vaccination with BCG has been shown to increase the efficacy of subsequent specific influenza vaccination [16]. However, in a study of a Mexican population, it was shown that genetic variants are involved in the presentation of severe forms of tuberculosis, even in vaccinated people [17]. Therefore, BCG vaccination may represent a synergistic factor for subsequent anti-SARS-CoV-2 vaccination. This research aimed to analyze the effect of BCG revaccination and genetic variants related to severe presentations of tuberculosis on subsequent anti-SARS-CoV-2 vaccination in exposed medical personnel.

## 2. Materials and Methods

### 2.1. Experimental Design and Participants

A prospective, double-blind, randomized study was conducted to evaluate the effect of BCG revaccination on subsequent anti-SARS-CoV-2 vaccination. Participants of the study were medical personnel, nurses, and support staff assigned to the General Hospital of Mexico, Mexico City, who had been assigned to treat patients with COVID-19 in the different facilities equipped for this purpose. The General Hospital of Mexico is a national referral center for multiple diseases and treats a large number of people from all over the country, with medical personnel from different specialties. As an institutional strategy against COVID-19, a bi-monthly polymerase chain reaction (PCR) monitoring protocol was established for all personnel of the hospital; for the personnel that participated in this study, the monitoring was carried out monthly. In addition, telephone surveys were conducted to report any signs or symptoms compatible with SARS-CoV-2 infection. The study was designed to be carried out in three steps (Figure 1).

In the first step, anteroposterior chest radiography and the Mantoux test through the inoculation of purified protein derivative (PPD) (TUBERSOL™, SANOFI PASTEUR) skin test were performed on each of the prospective participants. Those with negative results in both diagnostic tests were considered suitable. Other exclusion criteria included pregnant personnel, and those diagnosed with degenerative diseases, immunosuppression, or infections at the beginning of the protocol. In the second step, two groups of participants were integrated: one group received the BCG vaccine (PASTEUR MÉRIUX CONNAUGHT™) (0.1 mL previously reconstituted, intradermally with syringes and special needles on the outer side of the left arm), and the second group received an organoleptically similar placebo in the same amount and site. At this time, any adverse reaction was recorded, as well as anthropometric data. Thirty days after vaccination, an 8 mL sample of blood without anticoagulant was taken to measure the concentration of serum cytokines. In the third step, thirty days after administration of the BCG vaccination or placebo, participants in both groups received the first dose of the anti-SARS-CoV-2 vaccine (BNT_162b2_, Pfizer–BioNTech) (0.3 mL intramuscularly in the deltoid of the left arm), and 21 days later the second dose was given at the same condition. Finally, 30 days after the second dose of the anti-SARS-CoV-2 vaccine, two tubes of 8 mL peripheral blood were drawn, one without anticoagulant to determine the serum cytokine concentration, and the other with EDTA as an anticoagulant for DNA extraction. The local institutional review board approved the study protocol (approval code number: DI/20/601/04/29). Informed consent was given to all patients to participate in the study.

### 2.2. Cytokine Immunoassays

The serum was separated from the whole blood sample without anticoagulant by centrifugation and stored at −20 °C. The levels of cytokines in the serum were determined using a commercially available ProcartaPlex Human Th1/Th2 panel (Thermo Fisher Scientific, Waltham, MA, USA, cat. EPX180-12165-901) which was used according to the manufacturer’s instructions. Interleukin (IL)-1β, IL-2, IL-4, IL-5, IL-6, IL-12p70, IL-13, IL-18, interferon (IFN)-γ, granulocyte–monocyte colony-stimulating factor (GM-CSF), and tumor necrosis factor (TNF)-α were quantified. Samples were homogenized and adjusted for further quantification in the Luminex^®^LABScan 100 (Luminex Corp. Austin, TX, USA) system. The cytokine concentrations were calculated using the standard curve generated by the five-parameter logistic regression method. Each point of the reference curve and each sample was analyzed in duplicate. In the samples where cytokines were undetectable, values of the detection limit were used for analyses. xPONENT 3.1 software (Luminex Corp. Austin, TX, USA) was used for data acquisition and analysis.

### 2.3. Neutralizing Antibody Quantification

The percentage of neutralization due to the presence of neutralizing antibodies against SARS-CoV-2 was determined using a specific enzyme immunoassay (SARS-CoV-2 Surrogate Virus Neutralization Test (sVNT) Kit, GeneScript^®^). The development and validation of this surrogate virus neutralization test assay have previously been reported [18]. This kit has also been validated for diagnosis with a 30% signal inhibition cut-off point for SARS-CoV-2 neutralizing antibody detection. The neutralization test was performed according to the manufacturer’s instructions. Briefly, negative and positive sample controls were diluted 1:10 with the sample dilution buffer and mixed with an equal volume of HRP conjugate (60 µL and 60 µL), and then were incubated at 37 °C for 30 min. Subsequently, 100 µL of this mixture was transferred to 96-well plates coated with recombinant hACE2 and incubated at 37 °C for 15 min. After the incubation, the supernatant was removed, and the plates were washed four times with the Wash Solution. Finally, 100 µL of tetramethylbenzidine (TMB) was added and incubated for 15 min at room temperature; the reaction was stopped with 50 µL of Stop Solution. The plates were read at 450 nm immediately afterwards. The inhibition rate was calculated with the following formula [19]:(1)$$\%\;\mathrm{Neutralization}=\left(1-\frac{\mathrm{OD}\;\mathrm{value}\;\mathrm{of}\;\mathrm{samples}}{\mathrm{OD}\;\mathrm{value}\;\mathrm{of}\;\mathrm{negative}\;\mathrm{control}}\right)s\;100\%$$

### 2.4. HLA Genotyping

Following the manufacturer’s instructions, DNA was extracted from peripheral blood cells using the commercial BDtract Genomic DNA isolation kit (Maxim Biotech, San Francisco, CA, USA). DNA concentrations were assessed by UV absorption spectrophotometry at a 260 nm wavelength via a NanoDrop device (Thermo Scientific, Wilmington, DE, USA). Contamination with organic compounds and proteins was determined by measuring the ratio absorbance at 260/280 nm. Samples were considered of good quality when the ratio was ~1.8.

Human leukocyte antigen (HLA)-DR low-resolution genotyping was performed through PCR with sequence-specific primers (SSP), using a Micro SSP^TM^ Generic HLA Class II DNA Typing Tray–DRB Only Lot 004 (One Lambda, SSP2LB, Canoga Park, CA USA). HLA-DRB typing includes alleles in the *DRB1*, *DRB3*, *DRB4*, and *DRB5* loci in 24 independent reactions. The DRB loci allele group covers *DRB1*01*, **03*, **04*, **07*, **08*, **11*, **12*, **13*, **14*, **15*, and **16*, as well as *DRB3*02*, *DRB3*03*, *DRB4*01*, and *DRB5*01* specificities (DR51, DR52, and DR53 serological equivalents).

Taq DNA Polymerase recombinant (Ref. EP0402. Thermo Scientific. Vilnius, Lithuania) was employed in all PCR amplifications. The amplicons were electrophoresed in 2% agarose gels containing 0.2 µg/mL ethidium bromide (cat. E1385, Sigma-Aldrich, Germany) for 40 min (30 V/cm), and amplified bands were visualized in a dual-intensity UV light transilluminator (UVP Inc. Upland, CA, USA), before being stored and evaluated in the Electrophoresis Documentation and Analysis System (EDAS 290) (Eastman Kodak, New Haven, CT, USA). A 100 bp ladder molecular weight marker (cat. CSL-MDNA-100BPH, Cleaver Scientific United Kingdom) was employed to facilitate the allele-specific identification of the weight bands.

### 2.5. Statistical Analysis

We describe data using the means ± SD or medians (IQR) as appropriate. Categorical variables are described with frequencies and percentages. We first evaluated if there were baseline differences between those receiving BCG and the placebo in the concentration of cytokines with t-tests or Wilcoxon rank-sum tests, as appropriate. Afterwards, we compared the levels of the same cytokines between the placebo group and BCG group 30 days after receiving a second dose of the SARS-CoV-2 vaccine, again using t-tests or Wilcoxon rank-sum tests, according to the distributions of the variables. We also evaluated if there were significant differences in the concentrations of cytokines after the administration of SARS-CoV-2 vaccine within each group—this comparison was performed with the signed-rank test or paired t-test as appropriate. To compare the concentrations of neutralizing antibodies against SARS-CoV-2, the Wilcoxon signed-rank test was performed. All analysis was two-sided, and alpha was set at 5%: Stata 14.2 was used for all analyses.

Allele frequencies were determined by the direct counting of alleles identified in each participant, and the differences between groups were evaluated by the determination and comparison of the allele frequencies. To adjust for the confounding effect of the HLA alleles over the concentration of neutralizing antibodies, we elaborated multivariate linear regression analysis to include the following variables: BCG status and HLA status (homozygous or heterozygous) to estimate the adjusted β and its 95% CI to compare with the crude β of the corresponding variables of the univariate linear regression analysis. Statistical significance was assessed using Epi Info 7.1.4.0 [20] statistical software, considering the χ^2^ value to compare case and control groups. The results were considered significant when the *p*-value was <0.05 and corrected by the Yates test.

## 3. Results

### 3.1. Design and Participants

In step 1, sixty participants from the Hospital General de México were enrolled in the protocol. All participants were negative on the Mantoux test and did not present suggestive features of pulmonary disease due to tuberculosis based in the chest radiography. Most of the participants were women (75%), with a median body mass index (BMI) of 27.15 kg/m^2^ (24.45–30.35). The BCG vaccine is universally applicable in Mexico; all participants had a history of previous immunization in childhood (in the first thirty days of life), verified through the national vaccination records. Physicians had been treating COVID-19 patients for 25 h/week (Table 1).

Step two: at this time, two groups of 30 participants were randomly integrated. The first group received the BCG vaccine, whereas the other received the placebo. There were no statistical differences between groups regarding gender, age, BMI, or responses to the Mantoux test (Table 2).

At the end of this step, none of the participants were infected with SARS-CoV-2. Step three: both groups of participants received the anti-SARS-CoV-2 vaccine as part of the national vaccination program for occupationally exposed personnel. At the end of this period, several members of each group tested positive for infection with SARS-CoV-2 (Placebo–anti-SARS-CoV-2: 9 infected. BCG–anti-SARS-CoV-2: 14 infected); however, only two participants in the BCG–anti-SARS-CoV-2 group required hospitalization due to the presentation of severe clinical symptoms (Table 2).

### 3.2. Revaccination with BCG with Subsequent Vaccination against SARS-CoV-2 Induces Higher Serum Cytokine Concentrations

To evaluate the immune response induced in the vaccination protocol, the concentrations of several cytokines were determined in the serum at the end of steps 2 and 3. When comparing the serum levels of cytokines 30 days after the application of the BCG vaccine or a placebo, only the levels of IL-1β and IL-4 were significantly higher in the group that received the BCG vaccine (Appendix A). However, when comparing the serum concentrations of cytokines 30 days after the second dose of the anti-SARS-CoV-2 vaccine, in nine of eleven cytokines (IL-1β, IL-4, IL-6, IL-12p70, IL-13, IL-18, IFN-γ, GM-CSF, and TNF-α), we observed higher serum levels in the BCG–anti-SARS-CoV-2 group compared with the Placebo–anti-SARS-CoV-2 group (Figure 2).

On the other hand, when comparing the cytokines in the group that received the placebo before and after having received the anti-SARS-CoV-2 vaccine, it was observed that IL-1β, IL-2, IL-5, IL-18, IFN-γ, and TNF-α showed significant increases (Appendix A). In the group that received the BCG vaccine, the eleven cytokines analyzed showed significant increases (Appendix A).

### 3.3. Neutralizing Anti-SARS-CoV-2 Antibodies Were Higher in the Group of Participants with Revaccination with the BCG and Anti-SARS-CoV-2 Vaccines

To determine the effect of the vaccination protocol on the humoral immune response, blood was drawn from the participants, and neutralizing anti-SARS-CoV-2 antibodies were determined. For this purpose, we used a system that determined the percentage of neutralization as a reflection of the serum concentration of specific anti-SARS-CoV-2 neutralizing antibodies. At the end of stage 3, our result showed that the participants in the group with the combination of BCG and anti-SARS-CoV-2 vaccines had significantly higher neutralization percentages (89.57% [63.43–98.27%]), than the group of participants with the placebo–COVID-19 combination (63.49% [13.45–80.42%]) (Figure 3).

### 3.4. HLA Genotyping

To evaluate the frequency of HLA present in the participants of the different groups, DNA was purified from peripheral blood cells and used in the assay. Table 3 shows the HLA-DRB1 allele frequencies of Placebo–anti-SARS-CoV-2 and BCG–anti-SARS-CoV-2 groups.

We identified nine alleles in the BCG–anti-SARS-CoV-2 group and twelve in the Placebo–anti-SARS-CoV-2 group. In the BCG–anti-SARS-CoV-2 group, five alleles had a frequency of over 10%, and three of the alleles were responsible for over 50% of the variability (DRB1*04, DRB1*07, and DRB1*08); however, in the Placebo–anti-SARS-CoV-2 group, only the HLA-DRB1*04 allele reached 45% frequency. Interestingly, we observed a much higher frequency of the DRB1*04 allele in the Placebo–anti-SARS-CoV-2 group (45%) than in the BCG–anti-SARS-CoV-2 group (23%), which was statistically significant after Yates’ correction (*p* = 0.025). On the other hand, Table 4 shows univariate and multivariate regression analysis results of the possible confounding effect of HLA-DRB1*04 on the percentage of neutralizing antibodies. Our data showed no confounding effects (*p* > 0.05).

## 4. Discussion

Humankind is currently facing a novel pandemic that has affected people’s survival and quality of life around the world. The main alternative to controlling the SARS-CoV-2 pandemic is the generation of immunity through the application of vaccines; however, multiple factors influence the immune response induced by these immunogens, some in favor, and others that limit the generation of immunity. This study was designed to explore the effect of BCG revaccination on medical personnel occupationally exposed and subsequently vaccinated against SARS-CoV-2. Our data show that this combination synergizes the immune effect.

Although there is scant information on how the BCG vaccine could modify the development of infection from SARS-CoV-2, the number of reports is increasing rapidly [21]; however, some are controversial [22] and most of these studies propose training the immune system for subsequent contact with the novel coronavirus [23]. However, most of these hypotheses are based on statistical relationships in which it is intended to evaluate the history of vaccination in childhood, and parameters such as morbidity and mortality due to COVID-19 [24]. As far we know, the present study is the first to explore BCG revaccination aimed explicitly to modulate the subsequent vaccine response against SARS-CoV-2.

Interestingly, thirty days after the application of the BCG vaccine, the quantified cytokine levels did not show significant differences compared to the group of participants who received the placebo. This finding corroborates what was previously observed by Parlane et al., who developed a protocol of revaccination with the BCG vaccine in cattle. These authors reported that IFN-γ levels were practically undetectable until five weeks after revaccination; however, after inoculation of the mycobacterium, the levels of this cytokine increased dramatically along with specific antibodies [25]. On the other hand, Angelidou et al. reported that different licensed BCG formulations varied in terms of the concentrations of lipids, proteins, and genetic material coming from the mycobacteria lysate. Although in initial stages, the concentrations of pro-inflammatory cytokines are modest and the activities of T lymphocytes are evident, which provide specific protection in the long term [26].

In the present study, we found that the anti-SARS-CoV-2 vaccine induces an immune response characterized by the release of cytokines such as IL-1β, IL-2, IL-4, IL-6, IFN-γ, and TNF-α. These findings coincide with a report presented by Bergamaschi et al., who observed that after the first dose of an anti-SARS-CoV-2 vaccine (BNT162b2, Pfizer–BioNTech), levels of IL-15 and IFN-γ are increased, and after the second dose, the immune response is complemented by TNF-α and IL-6 [27]. Furthermore, our results showed that prior BCG vaccination increases this effect. This synergistic effect supports previous reports suggesting a beneficial effect of tuberculosis vaccination in the reduction in SARS-CoV-2 infection [28].

The integrated immune response induced by the cytokines mentioned above, such as IL-2, would indicate a direct proliferative stimulus to the different populations of lymphocytes [29]. At the same time, IL-1β, IL-4, IL-6, IFN-γ, GM-CSF, and TNF-α activate inflammatory effector mechanisms typical of a viral infection [30]. The integration of cytokine signals could reflect a better sensitization and efficient induction of immunological memory. Indeed, the synergistic immune response between the BCG vaccine and anti-COVID-19 vaccine was reflected in a significant increase in the levels of neutralizing antibodies in the group of participants who received both vaccine stimuli, compared with the group that received only the anti-SARS-CoV-2 vaccine.

At the end of the protocol, two of the participants in the group that received the BCG and the anti-SARS-CoV-2 vaccine developed severe signs of infection and had to be hospitalized, although this did not represent a statistically significant difference between the groups (Table 2). It is possible that the development of the phenomenon known as a cytokine storm, previously described in patients with severe infection [31], induced a great number of outliers, which was evident in the determination of cytokines in this group and is shown in Figure 2.

The observed synergistic phenomenon of BCG coincided with that described by Leentjens et al., who developed a model very similar to that used in this study; these researchers immunized a group of healthy volunteers with the BCG vaccine, who were later vaccinated against influenza A (H1N1). This immunization scheme resulted in higher levels of neutralizing antibodies compared to a control group that received only the influenza vaccine [16], supporting our results.

Although the proposal for the possible attenuating role of the BCG vaccine on morbidity and mortality caused by SARS-CoV-2 is based on statistical association studies [32], a possible cross-reaction in recognition of both pathogens should not be ignored because certain homology has been demonstrated between molecules present in both bacteria and the virus. Nouvo et al. demonstrated homology between SARS-CoV-2 and *M. bovis* proteins and proposed that they could be used to develop more efficient diagnostic techniques and explorations aimed at the induction of protective immunity [33]. The latter issue was addressed in a transcriptomic study published by Sharma [34], as well as an in silico analysis developed by Tomita et al. [35]. Based on our findings and previous reports, it is possible to hypothesize that increased concentrations of cytokines such as IL-2, IL-6, TNF-α, and IFN-γ could be involved in the synergistic effect, a phenomenon that could be heterologous or caused by homology between both pathogens.

On the other hand, when determining the percentage of neutralization due to the production of specific anti-SARS-CoV-2 neutralizing antibodies between both groups, the variability was notable. In the period from the application of the first dose of the anti-SARS-CoV-2 vaccine to the quantification of neutralizing antibodies thirty days after the second dose, some of the participants in both groups tested positive for COVID-19 infection. The results in Table 2 show that there was no significant difference between the groups; however, the effect of these infections could explain the variability observed in the quantification of neutralizing anti-SARS-CoV-2 antibodies. In experiments such as that carried out by Morales-Nuñez et al., it was reported that after the second dose of the anti-SARS-CoV-2 vaccine, 100% protection was observed in all inoculated [19]. Variability in vaccine responses could be associated with factors inherent to individuals. A report by Muller et al. indicated that the response induced by the anti-SARS-CoV-2 vaccine is dependent on age, and the production of neutralizing antibodies could take longer in older people [36], which could be associated with immunosenescence. This condition affects antigen presentation, the reduction in naive cells, effector mechanisms, and memory cells [37].

HLA is a molecule essential for activating the basal immune response in homeostasis, as well as in the context of infection, because it plays a central role in T cell activation. HLA class II (HLA-II) is expressed mainly on the cell surface of antigen-presenting cells, and there are three classic proteins called DR, DP, and DQ [38]. Different allelic variants potentially related to the inter-individual susceptibility and/or severity of COVID-19 have been reported [39]. Although no specific study has been developed describing genetic variants associated with COVID-19 in the Mexican population, one report has associated HLA-II alleles with the development of tuberculosis in this population [40]. To explore whether allelic variants were related to the defining variable of neutralizing antibody production, twelve alleles were screened. We found that the DRB1*04 allele exhibited a greater presence in the Placebo–anti-SARS-CoV-2 group. This allele has been reported to be associated with the development of multidrug-resistant tuberculosis [17]. Univariate and multivariate analyses were developed to verify whether this trait was related to the production of neutralizing antibodies, although our findings showed that this allele did not represent a confounding effect. However, this analysis showed that the BCG variable represented an increase of 23.80 after adjustment, with a confidence interval of 9.09–38.51 and *p* = 0.002.

The main strength of this study is that, unlike other reports where the statistical relationship between childhood vaccination of BCG and anti-SARS-CoV-2 has been explored, this study analyzed recent revaccination in tandem with its response to the anti-SARS-CoV-2 vaccine. On the other hand, this study had some limitations, such as the sample size; however, the significant differences observed open avenues for further research. The protocol focused on studying the humoral response; therefore, the other deficiency is that the cellular response was not analyzed. However, the findings regarding the cytokines could indicate the activation of different populations of lymphocytes.

## 5. Conclusions

Multiple factors can influence the effectiveness of vaccination against SARS-CoV-2; many of them are attributable to biological material, inherent to the individual, and depend on the strategy and the management of the vaccine. However, based on our findings, we can conclude that revaccination with BCG has a synergistic effect with an anti-COVID-19 vaccination, which represents a highly positive factor. This finding could represent an opportunity to achieve faster and more efficient protection against this novel coronavirus.

## Figures and Tables

**Figure 1 cells-10-03179-f001:**
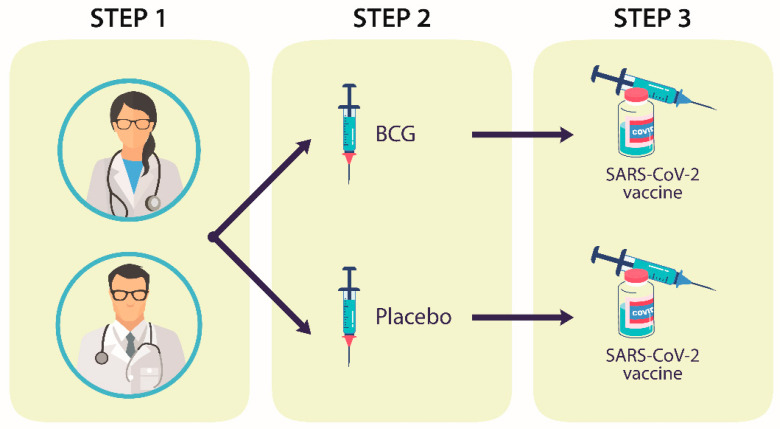
**Graphic diagram of the study. Step one:** sixty participants who had been occupationally exposed to SARS-CoV-2 were enrolled. **Step two**: two groups of 30 participants were randomly integrated; one group received a BCG vaccine, whereas the other group received a placebo. At the end of this step, the serum concentrations of pro-inflammatory cytokines were determined. **Step three**: both groups of participants received the Pfizer–BioNTech vaccine. At the end of this period, serum concentrations of pro-inflammatory cytokines and neutralizing antibodies were determined. Twelve HLA alleles were also genotyped.

**Figure 2 cells-10-03179-f002:**
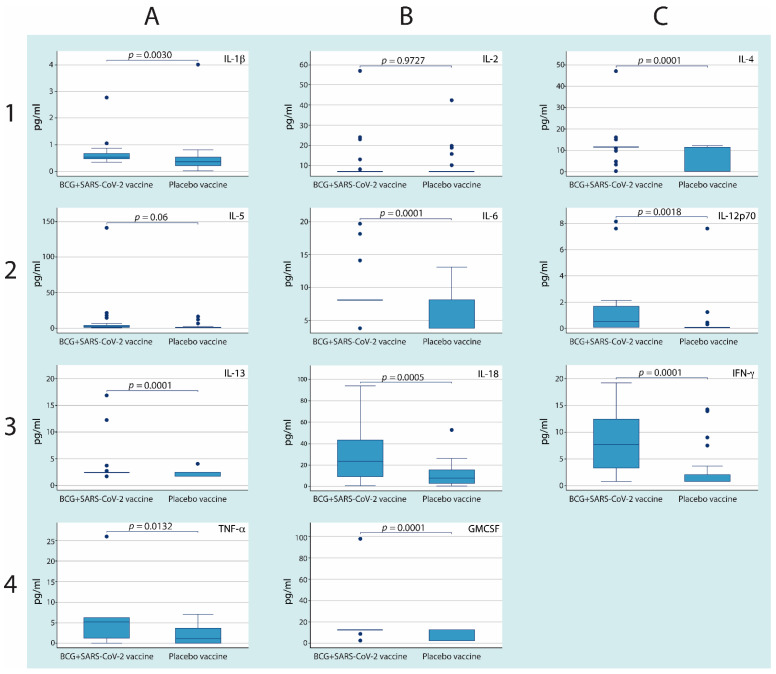
**BCG revaccination followed by SARS-CoV-2 immunization induces higher levels of serum cytokines.** Blood was drawn from each donor (*n*= 30 per group) thirty days after step 3, and cytokine levels were measured by ELISAs. Each graph shows a particular cytokine. Boxes represent the median ± IQR, whiskers extend to 95% confidence intervals, and individual dots represent outlier values. The lines show the *p*-values for comparisons between different groups.

**Figure 3 cells-10-03179-f003:**
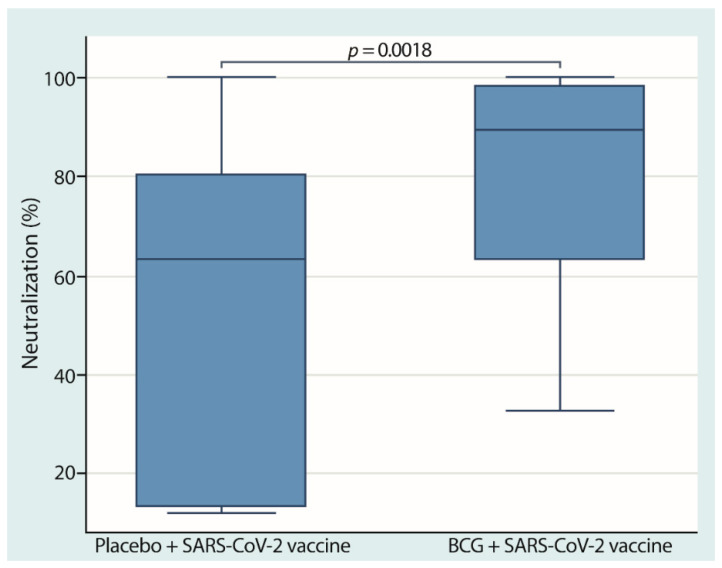
**Neutralizing antibody induction is higher in individuals previously revaccinated with BCG.** Blood was drawn thirty days after step three, and the levels of neutralizing antibodies against SARS-CoV-2 were determined by immunoassays. Boxes represent the median ± IQR, whiskers extend to 95% confidence intervals, and individual dots represent outlier values. The lines show the *p*-values for comparison between groups.

**Table 1 cells-10-03179-t001:** Demographic description of participants.

Variable	(*n* = 60)
Age, years	41 (30–50)
Female, *n* (%)	45 (75%)
Body mass index (BMI), kg/m^2^	27.15 (24.45–30.35)
BCG vaccine history, *n* (%)	60 (100%)
Mantoux test (PPD), mm	0 (0–5)
Time spent attending to patients with COVID-19, h/week	25

Continuous data are presented as medians (interquartile ranges) and frequencies of categorical data in percentages.

**Table 2 cells-10-03179-t002:** Comparison between experimental groups.

Variable	Placebo–anti-SARS-CoV-2	BCG–anti-SARS-CoV-2	*p*
Gender (*n*) F/M	24/6	21/9	0.371
Age (years)	42 (33–49)	38(28–54)	0.51
BMI (kg/m^2^)	24.65 (22.83–28.47)	25.23 (24.23–27.88)	0.6465
Mantoux test (PPD) (mm)	0 (0–5)	0 (0–6)	0.812
COVID-19 infection (without/with) (*n*)	21/9	16/14	0.184
Hospitalization (without/with) (*n*)	30/0	28/2	0.313

Continuous data are presented as medians (interquartile ranges) and frequencies of categorical data in percentage. Frequencies were compared with the chi-squared test; continuous data were compared using t-tests or Wilcoxon rank-sum tests according to the data distribution.

**Table 3 cells-10-03179-t003:** Allele frequency among Placebo–anti-SARS-CoV-2 and BCG–anti-SARS-CoV-2 subjects.

Allele	Placebo–anti-SARS-CoV-2		BCG–anti-SARS-CoV-2		
	*n* = 30	(%)	*n* = 26	(%)	*p*
DRB1*01	2	3.33	6	11.54	0.141
DRB1*03	1	1.67	0	0.00	NA
DRB1*04	27	45.00	12	23.08	0.025
DRB1*07	4	6.67	7	13.46	0.340
DRB1*08	8	13.33	8	15.38	0.969
DRB1*09	1	1.67	0	0.00	NA
DRB1*10	2	3.33	0	0.00	NA
DRB1*11	3	5.00	2	3.85	0.674
DRB1*13	3	5.00	5	9.62	0.468
DRB1*14:02	6	10.00	7	13.46	0.783
DRB1*15	1	1.67	4	7.69	0.181
DRB1*16	2	3.33	1	1.92	1.000

**Table 4 cells-10-03179-t004:** Univariate and multivariate regression analysis.

Variable	Crude β (95% CI)	*p*	Adjusted β (95% CI)	*p*
BCG	24.56 (10.73–38.39)	0.001	23.80 (9.09–38.51)	0.002
HLA-DRB1*04 heterozygous	−1.12 (−19.16–16.91)	0.9	0.08 (−17.42–17.6)	0.99
HLA-DR*04 homozygous	−11.30 (−30.14–7.54)	0.25	−3.83 (−22.9–15.23)	0.68

We present the crude β of BCG over the concentration of neutralizing anti-SARS-CoV-2 antibodies. BCG had a strong effect on the concentration of neutralizing antibodies. After adjusting for confounding effects of the HLA-DRB1*04 allele, no further confounding effects were found.

## Data Availability

Authors confirm the raw data to support this study's conclusions are included in the manuscript. The corresponding author will provide more information, upon reasonable request, to any qualified researcher.

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
