# Peer review of "Effect of BCG Revaccination on Occupationally Exposed Medical Personnel Vaccinated against SARS-CoV-2"

_cells, 2021, doi:10.3390/cells10113179_

Round 1

Reviewer 1 Report

The manuscript by Ramos-Martínez, et al., addresses the effect of BCG revaccination on circulating cytokines and anti-SARS-CoV-2 neutralizing antibodies after Pfizer COVID-19 vaccination. They used the term BCG revaccination as all children are vaccinated in Mexico with BCG under the national vaccination program. This is an interesting work that could be improved after solving some concerns.

MAJOR CONCERNS

  1. A relevant novelty of this study is the notion of “revaccination”. This concept is based on the administration of BCG in the childhood under the Mexican national program. But, is the protection conferred by this vaccine still present in the analyzed cohort? Did the authors test it by a Mantoux test? This could be a relevant limitation for this study.

  1. There is a statistical concern in the way the data are analyzed. The statistical design and tests performed allowed the comparison between two samples (t-test or Wilcoxon Sign Rank test). Then, in lines 214 and 215, for instance, the authors claim, “we observed higher serum levels in the BCG-anti-SARS-CoV-2 group, when 214 compared to the other three groups”. However, the statistical analysis does not allow this comparison.

  1. How did the SARS-CoV-2 infection after vaccination impact on the results? I mean, is there any differential behavior between SARS-CoV-2 infected or not participants? Could this account for the variability discussed in lines 328 – 337?

MINOR CONCERNS

  1. Lines 74-75. To the best of my knowledge, reference 73 does not study adaptive immunity or Th1 skewing. I would suggest to reformulate this sentence in order to strictly adhere it to the content of this work, the effect of BCG in monocytes and bone marrow HSPCs.

  1. Caption of figure 3 is not correct, please revise.

  1. Line 277, typo: panemic

Author Response

Dear Reviewer:

We deeply appreciate the review made in our manuscript, each of the recommendations and questions has been addressed; and without a doubt these observations turn our work in a better one. The modifications have been highlighted in red in the text, in this way we hope the revision will be a little easier.

The manuscript by Ramos-Martínez, et al., addresses the effect of BCG revaccination on circulating cytokines and anti-SARS-CoV-2 neutralizing antibodies after Pfizer COVID-19 vaccination. They used the term BCG revaccination as all children are vaccinated in Mexico with BCG under the national vaccination program. This is an interesting work that could be improved after solving some concerns.

MAJOR CONCERNS

  1. A relevant novelty of this study is the notion of “revaccination”. This concept is based on the administration of BCG in the childhood under the Mexican national program. But, is the protection conferred by this vaccine still present in the analyzed cohort? Did the authors test it by a Mantoux test? This could be a relevant limitation for this study.

Response:

          In Mexico, unfortunately the prevalence of pulmonary tuberculosis continues at high levels in the population, so the application of the BCG vaccine in childhood is part of a national program, all participants in the protocol had this vaccination history and the immunological history was evaluated. by means of the Mantoux test through the application of a purified protein derivative, in the previous text it was only referred to as PPD; however, in this new version we place the clarification in the text. In table 2, we show the comparison in this parameter and no statistically significant differences were found.

  1. There is a statistical concern in the way the data are analyzed. The statistical design and tests performed allowed the comparison between two samples (t-test or Wilcoxon Sign Rank test). Then, in lines 214 and 215, for instance, the authors claim, “we observed higher serum levels in the BCG-anti-SARS-CoV-2 group, when 214 compared to the other three groups”. However, the statistical analysis does not allow this comparison.

Response:

Reviewer is right, we appreciate the comment and we have made the corrections to better describe the data.

  1. How did the SARS-CoV-2 infection after vaccination impact on the results? I mean, is there any differential behavior between SARS-CoV-2 infected or not participants? Could this account for the variability discussed in lines 328 – 337?

Response:

In the first part of the protocol, participants in both groups were not positive for SARS-Cov-2 infection; however, in the final part of the protocol, participants from both groups were positive. The possibility that this factor was a determining factor in the variability observed when quantifying neutralizing antibodies was added to the discussion of the manuscript.

MINOR CONCERNS

  1. Lines 74-75. To the best of my knowledge, reference 73 does not study adaptive immunity or Th1 skewing. I would suggest to reformulate this sentence in order to strictly adhere it to the content of this work, the effect of BCG in monocytes and bone marrow HSPCs.

Response:

 The reference indicated was changed to one that adhered to the main argument of the manuscript.

  1. Caption of figure 3 is not correct, please revise.

Response:

The caption of figure 3 was changed.

  1. Line 277, typo: panemic

Response:

The spelling of this word was checked.

Reviewer 2 Report

To Authors:

Since the point of the paper is to determine if BCG potentiates a greater anti-SARS-CoV2 response, the paper would have been made much stronger if a third group had been included where the subjects were reinoculated with a different childhood vaccine 30 days before beginning the anti-SARS-CoV2 vaccines. This would help to determine whether any revaccination, or the BCG vaccine specifically, is potentiating a greater anti-SARS-CoV2 response.  Without this group, the methodology still could yield interesting information however the paper contains problems which unfortunately make it unsuitable for publication.

It is well-known that SARS-CoV2 infection can present with very mild symptoms, so it’s absolutely critical to test the first blood draw for SARS-CoV2 neutralizing antibodies.  Otherwise, you can’t say that the subjects were never infected before this blood draw, and it makes it impossible to attribute the greater antibody response in the second blood draw to the BCG revaccination. 

The authors state in the results that “At the end of this step [BCG revaccination vs placebo], none of the participants were infected by SARS-CoV-2.” (Line 203), what test was done to determine this? And does it mean they were not currently infected or never infected? This test is nowhere in the methods.  Since this is a sample of occupationally exposed subjects, the likelihood that none were ever infected seems low. By the second blood draw 14 subjects in the BCG group were infected with COVID-19 vs 9 in the placebo group (again, it’s not stated how this was determined).  This is a 55% increase in COVID-19 infections.  It’s certainly possible that the increase in cytokines and neutralizing antibodies seen (Figure 3) is entirely due to the increased rate of infection in this group, especially if they had an active infection at the time of the blood draw.  When they were infected in relation to the blood draw is also critical information since cytokine concentrations will change over the course of infection.  Moreover, this difference in infection isn’t mentioned at all in the discussion, it seems it wasn’t even noticed by the authors since they just state “several members of each group were positive for infection by SARS-CoV-2" (line 206) without quantifying how many members were infected. As the increased neutralizing antibodies in the BCG group vs placebo seems to be the crux of the paper, these problems present a fatal flaw.

Further to the problems with COVID-19 infections, there are statistical issues that make it difficult to even know that there are real differences in cytokine concentrations. It’s probably poor practice to alternate between parametric and non-parametric statistical tests within the same analysis (eg the concentration of cytokines between BCG vs placebo), if some of your groups are non-normally distributed, you should just stick with non-parametric tests for the whole analysis.  However, it is entirely inappropriate to test 11 different pairs with only t-tests or Wilcoxon Rank Sum tests without adjusting alpha to account for multiple comparisons. This is clearly a situation where a one-way ANOVA or Kruskal-Wallis test should be used.

More minor issues:

Approximately half of the intro is taken up by very well-trod, very general SARS-COV2/COVID info. It needs more focus on the rationale for needing an improved response to current vaccines and why BCG was chosen to attempt to potentiate a current coronavirus vaccine.  It also should include information about the alleles genotyped and why they were tested.  This isn’t even brought up until the discussion.

Figure 2: The extent of the boxes and whiskers are not explained.  Ie) do they represent mean, SEM and whiskers extend to 95% confidence interval?  There are many ways to represent these plots, so the parameters must be stated to help the viewer interpret them.  The labels on the axes and for p-values are nearly illegible because they are provided at such low size and resolution.

Line 135: RBD is not expanded.

Line 186: Something isn’t printing correctly in the PDF, there are spirals instead of characters.

Throughout there are basic proof-reading errors in need of correction, eg) “After each vaccine round of vaccination” line 27.

Author Response

We deeply appreciate the review made in our manuscript, each of the recommendations and questions has been addressed; and without a doubt these observations turn our work in a better one. The modifications have been highlighted in red in the text, in this way we hope the revision will be a little easier.

Since the point of the paper is to determine if BCG potentiates a greater anti-SARS-CoV2 response, the paper would have been made much stronger if a third group had been included where the subjects were reinoculated with a different childhood vaccine 30 days before beginning the anti-SARS-CoV2 vaccines. This would help to determine whether any revaccination, or the BCG vaccine specifically, is potentiating a greater anti-SARS-CoV2 response.  Without this group, the methodology still could yield interesting information however the paper contains problems which unfortunately make it unsuitable for publication.

It is well-known that SARS-CoV2 infection can present with very mild symptoms, so it’s absolutely critical to test the first blood draw for SARS-CoV2 neutralizing antibodies.  Otherwise, you can’t say that the subjects were never infected before this blood draw, and it makes it impossible to attribute the greater antibody response in the second blood draw to the BCG revaccination. 

The authors state in the results that “At the end of this step [BCG revaccination vs placebo], none of the participants were infected by SARS-CoV-2.” (Line 203), what test was done to determine this? And does it mean they were not currently infected or never infected? This test is nowhere in the methods.  Since this is a sample of occupationally exposed subjects, the likelihood that none were ever infected seems low. By the second blood draw 14 subjects in the BCG group were infected with COVID-19 vs 9 in the placebo group (again, it’s not stated how this was determined).  This is a 55% increase in COVID-19 infections.  It’s certainly possible that the increase in cytokines and neutralizing antibodies seen (Figure 3) is entirely due to the increased rate of infection in this group, especially if they had an active infection at the time of the blood draw.  When they were infected in relation to the blood draw is also critical information since cytokine concentrations will change over the course of infection.  Moreover, this difference in infection isn’t mentioned at all in the discussion, it seems it wasn’t even noticed by the authors since they just state “several members of each group were positive for infection by SARS-CoV-2" (line 206) without quantifying how many members were infected. As the increased neutralizing antibodies in the BCG group vs placebo seems to be the crux of the paper, these problems present a fatal flaw.

Response:

Each of the previous observations was addressed, and this can be seen in the new version of our text. The main doubt regarding the method of surveillance and detection of participants infected with SARS-Cov-2 was clarified in the methodology sections.

Further to the problems with COVID-19 infections, there are statistical issues that make it difficult to even know that there are real differences in cytokine concentrations. It’s probably poor practice to alternate between parametric and non-parametric statistical tests within the same analysis (eg the concentration of cytokines between BCG vs placebo), if some of your groups are non-normally distributed, you should just stick with non-parametric tests for the whole analysis.  However, it is entirely inappropriate to test 11 different pairs with only t-tests or Wilcoxon Rank Sum tests without adjusting alpha to account for multiple comparisons. This is clearly a situation where a one-way ANOVA or Kruskal-Wallis test should be used.

Response:

There must be, congruence in the description of the data and the analysis performed. This is a validity issue, if data has a non parametric distribution it must be described with medians and IQR. The analysis of the data must also follow this rules for the validity of the comparisons. We do not agree that have to use an ANOVA or a Kruskal-Wallis test, because we are just performing comparison of two groups in all performed analysis. In any instance we are performing the comparison of two groups. We agree that in the previous version of our manuscript, we presented the data in an incorrect way and as commented by another reviewer, some confusion in this issue was evident. In the current version of the manuscript, we have corrected this problem.

More minor issues:

Approximately half of the intro is taken up by very well-trod, very general SARS-COV2/COVID info. It needs more focus on the rationale for needing an improved response to current vaccines and why BCG was chosen to attempt to potentiate a current coronavirus vaccine.  It also should include information about the alleles genotyped and why they were tested.  This isn’t even brought up until the discussion.

Figure 2: The extent of the boxes and whiskers are not explained.  Ie) do they represent mean, SEM and whiskers extend to 95% confidence interval?  There are many ways to represent these plots, so the parameters must be stated to help the viewer interpret them.  The labels on the axes and for p-values are nearly illegible because they are provided at such low size and resolution.

Line 135: RBD is not expanded.

Line 186: Something isn’t printing correctly in the PDF, there are spirals instead of characters.

Throughout there are basic proof-reading errors in need of correction, eg) “After each vaccine round of vaccination” line 27.

Response:

Each of the previous minor issues were addressed and the modifications were incorporated into the new text that is presented.

Reviewer 3 Report

This manuscript report the effect of BCG vaccine for the pretreatment of COVID-19 vaccination by the well-designed protocol.  From the beginning the COVID-19 pandemic, the effect was speculated because of the lower number of COVID-19 patients in BCG vaccine recommended countries compared with not-fully recommended countries.  Therefore, I think this work is valuable and should be published in some paper.  Before that the authors should answer or add description against comments and question as follows:

1) Figure 2 is too small and p-values cannot be seen.  Also, no explanation of difference between dot and square.

2) Also in Figure 2, BCG and Placebo mean before the COVID-19 vaccination?  If so, it looks no difference in all cytokines. I think BCG vaccine contains many compounds including adjuvants, immune-stimulating reagents, therefore there should be some difference after 30 days.  Please explain.

3) The figure legend of Figure 3 is not appropriate.  It is for Figure 2.

4) In Table 2, what means of p value for Gender, Age, Covid-19 infection and Hospitalization?  Also Covid -19 infection 21/9 means 9 in 30 people, 16/14 means 14 in 30?  Hospitalization 30/0 means 0 in 30, 28/2 means 2 in 30? 

4) The inhibition (probably the neutralizing antibody, Figure 3) of BCG+Anti-SARS-CoV-2 was higher than that of Placebo+Anti-SARS-CoV-2.  It looks like the working condition of people of BCG-Anti-SARS-CoV-2 group may not be equal to people of Placebo- Anti-SARS-CoV-2 group. 

Author Response

We deeply appreciate the review made in our manuscript, each of the recommendations and questions has been addressed; and without a doubt these observations turn our work in a better one. The modifications have been highlighted in red in the text, in this way we hope the revision will be a little easier.

This manuscript report the effect of BCG vaccine for the pretreatment of COVID-19 vaccination by the well-designed protocol.  From the beginning the COVID-19 pandemic, the effect was speculated because of the lower number of COVID-19 patients in BCG vaccine recommended countries compared with not-fully recommended countries.  Therefore, I think this work is valuable and should be published in some paper.  Before that the authors should answer or add description against comments and question as follows:

  • Figure 2 is too small and p-values cannot be seen.  Also, no explanation of difference between dot and square.

Response:

The figure was modified for your better appreciation, in addition the explanation was modified. Each box represents a different cytokine and in each box the dots represent outliers.

  • Also in Figure 2, BCG and Placebo mean before the COVID-19 vaccination?  If so, it looks no difference in all cytokines. I think BCG vaccine contains many compounds including adjuvants, immune-stimulating reagents, therefore there should be some difference after 30 days.  Please explain.

Response:

In the discussion, a paragraph was added in which we touched on the observation made about the levels of cytokines induced after the application of the BCG vaccine.

3) The figure legend of Figure 3 is not appropriate.  It is for Figure 2.

Response:

The caption of figure 3 was changed.

  • ¿In Table 2, what means of p value for Gender, Age, Covid-19 infection and Hospitalization?  Also Covid -19 infection 21/9 means 9 in 30 people, 16/14 means 14 in 30?  Hospitalization 30/0 means 0 in 30, 28/2 means 2 in 30? 

Response:

The p value of the fourth column in Table 2 indicates the statistical probability value, according to the consideration that values equal to or less than 0.05 would indicate a difference between both groups of participants. In column one the clarification has been placed for a better understanding of which participants were infected and hospitalized.

  • The inhibition (probably the neutralizing antibody, Figure 3) of BCG+Anti-SARS-CoV-2 was higher than that of Placebo+Anti-SARS-CoV-2.  It looks like the working condition of people of BCG-Anti-SARS-CoV-2 group may not be equal to people of Placebo- Anti-SARS-CoV-2 group. 

Response:

 According to the inclusion and exclusion criteria, all participants in the protocol had similar conditions of exposure to the disease due to work conditions, and in the comparisons in gender, age and body mass index, no differences were observed that could influence the the result of the protocol.

Round 2

Reviewer 2 Report

The information regarding surveillance monitoring for COVID infection by PCR greatly aids in interpreting the data. While the fact that more subjects in the BCG group ended up with COVID by coincidence is a confounding factor, it is at least now acknowledged. 

I still think there are more than two groups being compared, so they cannot be considered independent (eg. Placebo vs BCG and Placebo vs Placebo+SARS-CoV2 makes three groups even though they are broken into different analyses and one analysis is not paired and the other is), but if I’m alone in this thinking, as I’m not an expert in statistics, I will concede. 

Since Figures 2 and 3 are box and whisker plots, please indicate the extent of the boxes and whiskers in the legends.  eg) “Boxes represent median ± IQR, whiskers extend to 95% confidence interval, individual dots represent outliers”; or whatever criteria was used. This is needed info for the reader since different criteria can be used in these types of plots ie) the whiskers could represent 1.5 x IQR.   

The paper still needs editing for grammar/spelling and I think the intro would be improved by moving the rationale for genotyping to the intro.

Author Response

Dear Reviewer:

We appreciate the time dedicated to the new revision of our manuscript, we have endeavored to answer each of the questions posed and we hope that they will find a better work and more suitable for publication.

The information regarding surveillance monitoring for COVID infection by PCR greatly aids in interpreting the data. While the fact that more subjects in the BCG group ended up with COVID by coincidence is a confounding factor, it is at least now acknowledged. 

Response:

Thank you for this comment, we try to make the information clearer for a better understanding

I still think there are more than two groups being compared, so they cannot be considered independent (eg. Placebo vs BCG and Placebo vs Placebo+SARS-CoV2 makes three groups even though they are broken into different analyses and one analysis is not paired and the other is), but if I’m alone in this thinking, as I’m not an expert in statistics, I will concede. 

Response:

Thank you for this comment, we hope that with the modifications made the analysis carried out will be better appreciated.

Since Figures 2 and 3 are box and whisker plots, please indicate the extent of the boxes and whiskers in the legends.  eg) “Boxes represent median ± IQR, whiskers extend to 95% confidence interval, individual dots represent outliers”; or whatever criteria was used. This is needed info for the reader since different criteria can be used in these types of plots ie) the whiskers could represent 1.5 x IQR.   

Response:

Thanks for the suggestion, the explanatory legend was added in figures 2 and 3.

The paper still needs editing for grammar/spelling and I think the intro would be improved by moving the rationale for genotyping to the intro.

Response:

For the grammar and spelling edition, the correction service offered by the publisher was used; on the other hand, the rationale for genotification was added in the introduction.

Reviewer 3 Report

Regarding previous Figure 2, the idea using supplementary table is good to see the small differences.  Also the added description in the discussion is acceptable. However, the letters in Figure 2 are still too small.  Also the cutoff values for the outliers (dots) should be explained in the figure legend.

Even though no difference statistically (Table 2), two hospitalized people in the group of BCG-SARS CoV-2 vaccine had to be followed in regard to their symptoms.  If the two had a kind of cytokine storm, it might be bad effect of BCG stimulation. Author should add some description in the discussion.

Author Response

Dear Reviewer:

We appreciate the time dedicated to the new revision of our manuscript, we have endeavored to answer each of the questions posed and we hope that they will find a better work and more suitable for publication.

Regarding previous Figure 2, the idea using supplementary table is good to see the small differences.  Also the added description in the discussion is acceptable. However, the letters in Figure 2 are still too small.  Also the cutoff values for the outliers (dots) should be explained in the figure legend.

Response:

The font size was increased in figure 2 and a better graphic description was added.

Even though no difference statistically (Table 2), two hospitalized people in the group of BCG-SARS CoV-2 vaccine had to be followed in regard to their symptoms.  If the two had a kind of cytokine storm, it might be bad effect of BCG stimulation. Author should add some description in the discussion.

Response:

A paragraph was added regarding the suggestion made regarding the immunological status of the infected participants in the BCG-anti-SARS-Cov-2 group.